# Hamiltonian Simulation Using NVIDIA CUDA-Q

Anurag Ramesh[1][0009−0001−8360−8614], W. Michael Brown[2], Thomas Lubinski[34], and David E. Bernal Neira[1][0000−0002−8308−5016]

[1] Davidson School of Chemical Engineering, Purdue University, IN, USA
rames102@purdue.edu, dbernaln@purdue.edu
[2] NVIDIA Corporation, CA, USA
michbrown@nvidia.com
[3] QED-C Technical Advisory Committee - Standards, VA, USA
[4] Quantum Circuits Inc., CT, USA
tlubinski@quantumcircuits.com

**Abstract.** Simulating quantum systems is a foundational application in quantum computing, especially in fields like computational chemistry [1]. We present a scalable framework, the Quantum Economic Development Consortium (QED-C) Application-Oriented Benchmark Suite to evaluate the performance of quantum algorithms across hardware platforms. A key focus is leveraging NVIDIA CUDA-Q [2], a powerful GPU-accelerated platform for quantum-classical hybrid programming, to benchmark Hamiltonian simulation, Quantum Fourier Transform (QFT), and Phase Estimation (PE).

We simulate a range of physical systems within HamLib [3], including the transverse field Ising, Heisenberg, and Fermi-Hubbard models, as well as molecules such as $H_2$ and $B_2$ using Suzuki-Trotter evolution. Simulations were executed on NVIDIA GPU clusters, including the A100, H100, GH200, and GB200 systems, across Purdue University [4], Lawrence Berkeley National Laboratory (LBNL) [5] and in collaboration with NVIDIA. `CUDA-Q`'s `SpinOperator` formalism enabled emulation of circuits for up to 38 qubits on the LBNL cluster, with performance up to 3× faster than real quantum hardware. Strong scaling behavior is observed up to 32 GPUs, with execution times for some simulations reduced by more than 90%. For example, execution times for simulating a 33-qubit TFIM dropped from 41 s (1 GPU) to 2.8 s (32 GPUs).

Despite these gains, we observe classical HPC-like diminishing returns beyond 8 GPUs, due to inter-GPU communication bottlenecks. This impact is mitigated on the latest GB200 clusters that support extending the high-bandwidth NVLink GPU interconnect across multiple nodes. CUDA-Q proves especially effective for sampling-heavy workloads, offering near-linear scaling and improved parallel efficiency for PE and QFT as well. Our findings demonstrate that GPU-accelerated quantum simulation with CUDA-Q provides a robust, high-throughput alternative to noisy intermediate-scale quantum (NISQ) devices and paves the way for future kernel-level optimizations and distributed quantum computing strategies.

**Keywords:** Hamiltonian simulation · CUDA-Q · GPU acceleration · Trotterization · multi-GPU scaling

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
