# OpenReview forum: "Hamiltonian Simulation Using Nvidia CUDA-Q"
_purdue.edu/Purdue_University/PQAI/2025/Symposium — PQAI 2025 Poster_

### Official Review · Reviewer_Wkc8 · 2025-07-25
**A topic of interest to the community, but not a full paper, perhaps good for a tutorial session**

**Rating:** 6
**Confidence:** 5

**Review:**

The submitted paper consists of a single page abstract, only. However, the topic of the abstract is familiar to the reviewer and is likely of great interest to the community since the work is very practically focused --- digging deep into quantum benchmarking and GPU-accelerated simulation. Therefore, this submission seems very well suited either to a poster session or a dedicated tutorial session to give participants a chance to work hands on with the CUDA-Q simulator described in the submitted abstract.

---

### Official Review · Reviewer_zbxw · 2025-07-25
**Surely an interesting presentation**

**Rating:** 7
**Confidence:** 4

**Review:**

The submission describes a benchmarking framework for quantum algorithms, with an emphasis on Hamiltonian simulation using CUDA-Q. The abstract could be more focused (e.g., the title suggests Hamiltonian simulation is central, but the first paragraph introduces a broader benchmarking suite), but the overall contribution is clear.

The results presented, especially the scaling behavior across GPU clusters, are interesting. The application of CUDA-Q for large-scale simulations, including PE and QFT, is well motivated.

Comments to the Authors:
- Please clarify the focus of the talk early on. Is it about Hamiltonian simulation using CUDA-Q, or about the QED-C benchmarking framework more generally? The storyline would benefit from a more explicit structure from the beginning.
- The content is strong and the results are interesting. I would attend this talk.

Recommendation: Accept (Presentation)

---

### Decision · Program_Chairs · 2025-07-29

Reject